# ADVERSARIAL POLICY GRADIENT FOR ALTERNATING MARKOV GAMES

**Chao Gao**
cgao3@ualberta.ca

**Martin Müller**
mmueller@ualberta.ca

**Ryan Hayward**
hayward@ualberta.ca

## ABSTRACT

Policy gradient reinforcement learning has been applied to two-player alternate-turn zero-sum games, e.g., in AlphaGo, self-play REINFORCE was used to improve the neural net model after supervised learning. In this paper, we emphasize that two-player zero-sum games with alternating turns, which have been previously formulated as Alternating Markov Games (AMGs), are different from standard MDP because of their two-agent nature. We exploit the difference in associated Bellman equations, which leads to different policy iteration algorithms. As policy gradient method is a kind of generalized policy iteration, we show how these differences in policy iteration are reflected in policy gradient for AMGs. We formulate an adversarial policy gradient and discuss potential possibilities for developing better policy gradient methods other than self-play REINFORCE. The core idea is to estimate the *minimum* rather than the *mean* for the "critic". Experimental results on the game of Hex show the modified Monte Carlo policy gradient methods are able to learn better pure neural net policies than the REINFORCE variants. To apply learned neural weights to multiple board sizes Hex, we describe a board-size independent neural net architecture. We show that when combined with search, using a single neural net model, the resulting program consistently beats MoHex 2.0, the previous state-of-the-art computer Hex player, on board sizes from $9{\times}9$ to $13{\times}13$.

## 1 INTRODUCTION

Reinforcement Learning (RL) is a trial-and-error paradigm where an agent learns by interacting with an environment. The *prediction* problem in RL seeks to learn the true value function of a following policy, while the *control* problem aims to find an optimal policy that maximizes the long-term expected cumulative reward. The environment is assumed to be a Markov Decision Process (MDP) (Bellman, 1957). Compared to dynamic programming, RL methods have the advantage of being model-free which learn by treating the environment as a black box. Dynamical programming is theoretically fundamental to reinforcement learning (Bertsekas & Tsitsiklis, 1996), as almost all RL algorithms are a form of *generalized policy iteration*, which improve the policy while estimating a value function (Sutton & Barto, 2017).

Model-free reinforcement learning methods have been successfully applied to many domains, including robotics (Kober et al., 2013), Atari games (Mnih et al., 2015; 2016; Schaul et al., 2016), and two-player board games (Tesauro, 1995; Silver et al., 2016). Most RL algorithms fall into one of two categories: value fitting and policy gradient. The value-fitting algorithms try to optimize a value function through iteratively minimizing a sequence of Bellman inconsistencies of observed states or state-action pairs, examples are on-policy SARSA (Rummery & Niranjan, 1994) and off-policy Q-learning (Watkins & Dayan, 1992). An optimal deterministic policy is implicitly learned when the value function converges to optimal.

The primary disadvantage of value-fitting methods such as Q-learning (Watkins & Dayan, 1992; Mnih et al., 2015; Wang et al., 2016; Mnih et al., 2016) is that they might be unstable when interacting with function approximation (Bertsekas & Tsitsiklis, 1996; Sutton et al., 2000). To gain stable behavior, often extra heuristic techniques (Lin, 1992; Schaul et al., 2016) and extensive hyper-parameter tuning are required. By contrast, policy gradient methods explicitly represent a policy as a function of parameters: they learn through iteratively adjusting the parameterized policy by fol-

lowing estimation of gradients, thus converging to at least a local maximum (Peters & Bagnell, 2011). Policy gradients are applicable to continuous control (Silver et al., 2014; Schulman et al., 2015; 2017) or domains with large action space (Silver et al., 2016), whereas action-value learning methods often become infeasible (Lillicrap et al., 2016; Wang et al., 2017).

Both value and policy based reinforcement learning are well-studied in MDPs, where a single agent learns by exploiting a stationary environment. In this paper, we focus on policy gradient methods for two-player zero-sum games played in alternating turns, i.e., Alternating Markov Games (AMGs) (Littman, 1996). AMGs are a specialization of Stochastic Games (Shapley, 1953) but a generalization of MDPs by allowing exactly two players pursing diametrical goals. Many popular two-player games are of such kind, such as chess, checkers, backgammon, Hex and Go. The restriction of zero-sum in AMGs makes it possible to define a shared reward function such that one agent tries to maximize while the other agent tries to minimize it. Due to such a property, with a notion of *self-play* policy, standard reinforcement learning methods have been successfully applied to AMGs (Tesauro, 1995; Silver et al., 2016) by simply negating the reward signal of the opponent's turn.

In this paper, we reexamine the justifications of adapting standard reinforcement learning methods to AMGs. We begin by reviewing the fundamental differences in their corresponding Bellman equations, from which we show how the resulting policy iteration algorithms are disparate. We formulate an adversarial policy gradient objective specifically for AMGs, based on which we develop new policy gradient methods for AMGs. Specifically, we apply our approach to the game of Hex. We show that by modifying REINFORCE to estimate the *minimum* rather than the *mean* return of a self-play policy, stronger pure neural net players are obtained.

## 2 FINITE MDP

A finite MDP is a tuple $(\mathcal{S}, \mathcal{A}, \mathcal{R}, \mathcal{P}, \gamma)$ where $\mathcal{S}$ is finite set of states, $\mathcal{A}$ is a set of actions, $\mathcal{R} : \mathcal{S} \times \mathcal{A} \to \mathbb{R}$ is a reward function, and $\mathcal{P}$ defines the probabilistic transitions among states. An agent lives in an environment characterized by the Markov property, i.e., $\Pr(s_{t+1}|s_1, a_1, \ldots, s_t, a_t) = \Pr(s_{t+1}|s_t, a_t)$. The agent learns by receiving reward signals from interacting with the environment. The goal is to maximize the expected discounted cumulative reward: $\mathbb{E} \sum_{k=t}^{\infty} \gamma^{t-k} R_{t-k+1}$, where $\gamma$ is a discounting factor $0 < \gamma \leq 1$ that controls the contribution of long-term and short-term rewards; $\gamma = 1$ is only possible in episodic tasks.

The basis for reinforcement learning in MDP is a set of *Bellman Equations*. For a given policy $\pi$:

$$v_\pi(s) = \sum_a \pi(s,a) \sum_{s'} p(s'|s,a)(r(s,a,s') + \gamma v_\pi(s')), \tag{1}$$

where $p(s'|s,a)$ is the transition function and $r(s,a,s')$ is the reward of taking $a$ at $s$, leading to $s'$.

It is also popular to use an action-value function:

$$q_\pi(s,a) = \sum_{s'} p(s'|s,a)(r(s,a,s') + \gamma v_\pi(s')) \tag{2}$$

The above Bellman equations are the basis for *policy evaluation*, which computes the true value function for a given $\pi$. This corresponds to the *prediction* problem in RL. The *Optimal Bellman Equation* is a recursive relation for the optimal policy:

$$v_*(s) = \max_a \sum_{s'} p(s'|s,a)(r(s,a,s') + \gamma v_*(s'), \tag{3}$$

$$q_*(s,a) = \sum_{s'} p(s'|s,a)(r(s,a,s') + \gamma \max_{a'} q_*(s',a')). \tag{4}$$

A value iteration procedure derived by the above optimal Bellman equation converges to optimal (Bellman, 1957). Alternatively, it is also possible to derive a *policy iteration* (Howard, 1960; Bertsekas & Tsitsiklis, 1996) by combining Equation (1) and (3). In reinforcement learning, the precise model is usually assumed to be unknown. Learning typically interleaves *policy evaluation* and *policy improvement*, which is summarized as generalized policy iteration (Sutton & Barto, 2017).

# 3 ALTERNATING MARKOV GAMES

Alternating Markov Games (AMG) are a specialization of Stochastic Games with only two players: an AMG is a tuple $(\mathcal{S}_1, \mathcal{S}_2, \mathcal{A}_1, \mathcal{A}_2, \mathcal{R}, \mathcal{P}, \gamma)$ where $\mathcal{S}_1$ and $\mathcal{S}_2$ are respectively *this* and *other* agent's states, $\mathcal{A}_1$ and $\mathcal{A}_2$ are the actions at each player's states. MDPs can be viewed as a special case of AMGs by restricting $|\mathcal{S}_2| = 0$.

Since there are two players in AMGs, *policy evaluation* involves two policies, i.e., $\pi_1$ and $\pi_2$. Bellman equations for policy evaluation are:

$$\begin{cases} v_{\pi_1}(s) = \sum_a \pi_1(a|s) \sum_{s'} p(s'|s,a)(r(s,a,s') + \gamma v_{\pi_2}(s')), s \in \mathcal{S}_1 \text{ and } s' \in \mathcal{S}_2 \\ v_{\pi_2}(s) = \sum_a \pi_2(a|s) \sum_{s'} p(s'|s,a)(r(s,a,s') + \gamma v_{\pi_1}(s')), s \in \mathcal{S}_2 \text{ and } s' \in \mathcal{S}_1 \end{cases} \quad (5)$$

Action-value functions can also be defined:

$$\begin{cases} q_{\pi_1}(s,a) = \sum_{s'} p(s'|s,a)(r(s,a,s') + \gamma \sum_{a'} \pi_2(a'|s')q_{\pi_2}(s',a')), s \in \mathcal{S}_1 \text{ and } s' \in \mathcal{S}_2 \\ q_{\pi_2}(s,a) = \sum_{s'} p(s'|s,a)(r(s,a,s') + \gamma \sum_{a'} \pi_1(a'|s')q_{\pi_1}(s',a')), s \in \mathcal{S}_2 \text{ and } s' \in \mathcal{S}_1 \end{cases} \quad (6)$$

Let $\pi_1$ be the $max$ player and $\pi_2$ be the $min$ player. Assuming the $\pi_2$ is an optimal "counter policy" with respect to $\pi_1$, we may rewrite the above equation as:

$$q_{\pi_1}(s,a) = \sum_{s'} p(s'|s,a) \Bigg\{ r(s,a,s') + \gamma \min_{a'} \sum_{s''} p(s''|s',a') \\ \left[ r(s',a',s'') + \sum_{a''} \pi_1(a''|s'')q_{\pi_1}(s'',a'') \right] \Bigg\}, \quad (7)$$
$$\text{where } s \in \mathcal{S}_1, s' \in \mathcal{S}_2 \text{ and } s'' \in \mathcal{S}_1.$$

The replacement of $\pi_2$ with a min operator is caused by the observation that, since $\pi_1$ is fixed, the problem reduces to a single-agent MDP where an agent tries to minimize the recevied rewards.

The optimal Bellman equation (also implies Nash equilibrium) can be expressed as:

$$\begin{cases} v^*(s) = \max_a \sum_{s'} p(s'|s,a)(r(s,a,s') + \gamma v^*(s')), s \in \mathcal{S}_1 \text{ and } s' \in \mathcal{S}_2, \\ v^*(s) = \min_a \sum_{s'} p(s'|s,a)(r(s,a,s') + \gamma v^*(s')), s \in \mathcal{S}_2 \text{ and } s' \in \mathcal{S}_1, \end{cases} \quad (8)$$

assuming that states in $\mathcal{S}_1$ belongs to the $max$ player, the other one is the $min$ player. A *value iteration* algorithm according to this minimax recursion converges to optimal (Condon, 1992). However, because the *policy evaluation* in AMGs consists of two policies $\pi_1, \pi_2$, the policy iteration, which alternates between policy evaluation and policy improvement, could have the following four formats:

**Algo.1** Fix $\pi_1^t$, compute the optimal counter policy $\pi_2^t$, then fix $\pi_2^t$, compute the optimal counter policy $\pi_1^{t+1}$, alternating this procedure repeatedly,

**Algo.2** Policy evaluation with $\pi_1^t, \pi_2^t$, switch both $\pi_1^t$ and $\pi_2^t$ to greedy policies with respect to current state-value function, continue this procedure repeatedly,

**Algo.3** Policy evaluation with $\pi_1^t, \pi_2^t$, switch $\pi_1^t$ to greedy policy with respect to the current state-value function and then compute the optimal counter policy for $\pi_2^t$, continue this procedure repeatedly,

**Algo.4** Policy evaluation with $\pi_1^t, \pi_2^t$, switch $\pi_2^t$ to greedy policy with respect to the current state-value function and then compute the optimal counter policy for $\pi_1^t$, continue this procedure repeatedly.

Intuitively speaking, all of those procedures are somewhat sensible, and perhaps would yield practical success. However, Condon (1990) showed that oscillation could be a problem that prevents **Algo.1** and **Algo.2** from converging, only **Algo.3** and **Algo.4** are correct, guaranteed to converge in general (Hoffman & Karp, 1966). **Algo.3** and **Algo.4** are duals. Variants of **Algo.3** or **Algo.4**, such as only switching one node to greedy every iteration then optimize the other player's strategy, are also correct, although they may have slower convergence rate (Condon, 1990).

## 4 ADVERSARIAL POLICY GRADIENT

We first review policy gradient in single agent MDP. In MDP, the strength of a policy $\pi$ can be measured by (for brevity, we implicitly state that the policy is parameterized by $\theta$):

$$J(\pi) = \sum_{s \in \mathcal{S}} d^\pi(s) \sum_a \pi(a|s) q_\pi(s, a), \tag{9}$$

where $d^\pi(s)$ is a state distribution under $\pi$. The gradient of $J(\pi)$ is:

$$\nabla J(\pi) = \sum_{s \in \mathcal{S}} d^\pi(s) \sum_a \pi(a|s) \nabla \log \pi(a|s) q_\pi(s, a) \tag{10}$$

The above formula is the Policy Gradient Theorem in MDP (Sutton et al., 2000), which implies that the gradient of the strength of a policy can be estimated by sampling according to $\pi$. The requirements are 1) $\pi$ is differentiable, and 2) $q_\pi(s, a)$ can be estimated. This theorem can also be interpreted as a kind of generalized policy iteration, where the gradient ascent corresponds to policy improvement (Kakade, 2002), and $q_\pi(s, a)$ is obtained by some policy evaluation. Depending on how $q_\pi(s, a)$ is estimated, policy gradient algorithms can be categorized into two families: Monte carlo policy gradients that use Monte carlo to estimate $q_\pi$, e.g., REINFORCE (Williams, 1992) and actor-critic methods (Sutton et al., 2000; Wang et al., 2017) that use another parameter to approximate the action-value under $\pi$. Recent approaches (Gu et al., 2017) interpolate both.

Analogously, in AMGs, the joint strength for a pair of parameterized policies $\pi_1$ and $\pi_2$ may be defined as:

$$J(\pi_1, \pi_2) = \sum_s d^{\pi_1, \pi_2}(s) \sum_a \pi_1(a|s) q_{\pi_1}(s, a), \tag{11}$$

where $d^{\pi_1, \pi_2}(s)$ is the state-distribution given $\pi_1$ and $\pi_2$. A natural question is what is the gradient of $J(\pi_1, \pi_2)$ with respect to $\pi_1$ and $\pi_2$ respectively? One tempting derivation is to calculate the the gradient for both $\pi_1$ and $\pi_2$ simultaneously by treating the other policy as the "environment". Similar to the mutual greedy improvement in **Algo.2**, such a method tries to adapt based on the value function under current $\pi_1, \pi_2$, ignoring the fact that the opponent is a dynamic adversary who will also adapt its strategy for better counter-payoffs. Another possible algorithm is to fix $\pi_1$ and do a fix number of iterations to optimize $\pi_2$ by normal policy gradient as in MDP, and then fix $\pi_2$ for optimizing $\pi_1$, repeat such alternatively. However, this algorithm is an analog to **Algo.1**, which was shown to be non-convergent in general.

Following **Algo.3** and **Algo.4**, given the action-value function under $\pi_1, \pi_2$, a more reasonable approach for policy improvement is thus to switch $\pi_2$ to "greedy" (not necessary every node of $\pi_2$) and optimize $\pi_1$ by policy gradient. Therefore, assuming $\pi_1$ is the max player, we advocate the following objectives

$$\begin{cases} J^{\pi_1}(\pi_1, \pi_2) = \sum_s d^{\pi_1, \pi_2}(s) \sum_a \pi_1(a|s) \sum_{s'} p(s'|s, a) \left[ r(s, a, s') + \gamma \min_{a'} q_{\pi_2}(s', a') \right] \\ J^{\pi_2}(\pi_1, \pi_2) = \sum_s d^{\pi_1, \pi_2}(s) \sum_a \pi_2(a|s) \sum_{s'} p(s'|s, a) \left[ r(s, a, s') + \gamma \max_{a'} q_{\pi_1}(s', a') \right]. \end{cases} \tag{12}$$

Consequently, the gradients can be estimated by

$$\begin{cases} \nabla J^{\pi_1}(\pi_1, \pi_2) = \mathbb{E}_{\pi_1, \pi_2} \left[ \nabla \log \pi_1(a|s) \sum_{s'} p(s'|s, a)(r(s, a, s') + \gamma \min_{a'} q_{\pi_2}(s', a')) \right] \\ \nabla J^{\pi_2}(\pi_1, \pi_2) = \mathbb{E}_{\pi_1, \pi_2} \left[ \nabla \log \pi_2(a|s) \sum_{s'} p(s'|s, a)(r(s, a, s') + \gamma \max_{a'} q_{\pi_1}(s', a')) \right]. \end{cases} \tag{13}$$

The above formulation implies that, when computing the gradient for one policy, the other policy is simultaneously switched to "greedy". This joint-change forces the current player to adjust the action preferences according to the worst-case response of the opponent, which is desirable due to the adversarial nature of the game.

### 4.1 ADVERSARIAL MONTE CARLO POLICY GRADIENT

A straightforward implementation of Equation (12) is to obtain separate Monte carlo estimates for each next-action, and then apply the $\min$ or $\max$ operator. However, this may not be practically

feasible when the action space is large. We introduce a parameter $k$, by which only a subset of actions are considered in our adversarial Monte-Carlo policy gradient design.

As a minimal modification on self-play REINFORCE, the algorithm works as follows: after a batch of $n$ games is generated by a self-play policy with game results, for each game, a single state-action pair $(s, a)$ is sampled uniformly as a training example. However, instead of directly using the observed return $z$ in batch as the estimated action-value for $(s, a)$, we perform extra Monte carlo simulations to estimate the *minimum* return for $(s, a)$. In below are two methods we propose:

> **AMCPG-A:** Run $k$ self-play games from $(s, a)$ using self-play policy $\pi$, then take the minimum of the $k$ additional returns and $z$.

> **AMCGP-B:** Sample a $s'$ from $(s, a)$ according to state-action transition $p(\cdot|s, a)$, select top $k$ actions suggested by $\pi$, for each selected action, obtain an Monte carlo estimate using self-play policy $\pi$, then take the minimum of the $k$ additional returns and $z$.

Note that only "minimum" is used, because we assume the game result is with respect to the player to play at state $s$. It is easy to see that in **AMCPG-A**, if *mean* operator is used rather than *minimum*, a self-play REINFORCE variant is recovered. When $k = |\mathcal{A}(s')|$, **AMCPG-B** becomes a genuine implementation of Equation (12), but in this situation, even though each action-value's estimation is unbiased, bias could still be incurred when applying the $\min$ or $\max$. This is known as the "winner's curse" problem (Capen et al., 1971; Smith & Winkler, 2006).

We note that, as in MDPs, another possible direction is to implement Equation (12) in an adversarial actor-critic framework. The difference is that a $\min$ operator will be used when estimating the "critic". In practice, when neural net is used as a function approximator, the minimum action-value would be easier to obtain by employing an architecture with both policy and action-value outputs. More desirably, as in Q-learning (Fox et al., 2016), bias might be alleviated by certain soft $\min$ operators.

## 5 RELATED WORK

Games are studied by different disciplines, including game theory, reinforcement learning and computational complexity. Shapley (1953) introduced the notion of Stochastic Games, which is a multi-player framework that generalizes both Markov Decision Process (only one player) and repeated games (only one state). Condon (1990; 1992) initiated the study of *Simple Stochastic Games*, which are two-player games played on a directed graph with $min$, $max$ and restricted probabilistic transition nodes. While her concern was largely from a computational complexity perspective, Condon (1990) showed that several variants of Hoffman-Karp's algorithms are incorrect. Littman (1996) formulated the notion of Alternating Markov Games, which is more general than *Simple Stochastic Games* by removing the restriction in action sets and probabilistic transitions. Littman (1994) proposed a minimax-Q learning algorithm that is applicable to Alternating Markov Games as well as two-player zero-sum games played with matrix payoffs.

One of the most noticeable study that uses reinforcement learning to play AMGs is by Tesauro (1995), who trained multi-layer neural network to play backgammon. Tesauro's program takes advantage of the symmetric (due to alternating and zero-sum) property of AMGs and learns the optimal value function by "greedy" self-play, relying on the stochastic environment for exploration. It was conjectured that the smoothness of the value function is one major factor for the particular success of TD-Gammon (Tesauro, 1995).

The recent advances of reinforcement learning are arguably primarily due to the use of deep neural networks (LeCun et al., 2015) as a much more expressive function approximator. Deep neural nets have been applied the game of Go (Maddison et al., 2015; Tian & Zhu, 2016; Clark & Storkey, 2015), leading to super-human Go-playing system AlphaGo (Silver et al., 2016). AlphaGo consists of several components including supervised learning for move prediction, reinforcement learning and Monte Carlo Tree Search (MCTS) (Coulom, 2006). Most recent AlphaGo Zero uses a "heavy" self-play based on MCTS to improve its parameterized neural net (Silver et al., 2017b). A similar approach is Expert Iteration (Anthony et al., 2017). One drawback of such methods is their huge computational cost when using MCTS self-play for data generation. For example, when applied to Shogi and chess, 5000 TPUs were used (Silver et al., 2017a).

Policy gradient reinforcement learning is a conventional fast RL method that can be used to refined a neural net policy by pure neural net self-play. In AlphaGo (Silver et al., 2016), policy gradient RL is used to improve the neural network model obtained by supervised learning. The improved model was less useful in search, much because its strong bias towards a single move. However, due to its better playing strength and fast speed, it was used to generate training data for a value net (Silver et al., 2016), which can be integrated into Monte carlo tree search. The policy gradient employed by AlphaGo is a variant of self-play REINFORCE that resembles **Algo.2**. A practical innovation is to select an opponent from a pool of previous parameters, so as to increase the stability of the training. Policy gradient has also been applied to train a balanced policy (Silver & Tesauro, 2009); however, the requirement of an optimal supervision makes it less useful in practice (Huang et al., 2010).

Adversarial methods have also been adopted in MDP (Pinto et al., 2017), since errors may occur in simulated models, maximizing a worst-case return will generally produce more robust results. The alternating procedure proposed by Pinto et al. (2017) resembles **Algo.1**. The idea of adversarial learning is also used in generative models (Goodfellow et al., 2014), which leverage adversarial examples to train a more robust classifier.

# 6 EXPERIMENTS

In this section, we present experimental results of the proposed adversarial Monte carlo policy gradient algorithms of Section 4. We first introduce the game of Hex as a testbed, then present experimental results.

## 6.1 GAME OF HEX

The game of Hex is played on a $N \times N$ rhombus board, where black and white alternately place a stone on an unoccupied hexagonal cell. The goal is to connect two opposite sides of the board. Since its invention, the game of Hex has been an active research problem for mathematicians (Nash, 1952) and computer scientists (Shannon, 1953). Nash (1952) proved by strategy stealing argument that from the empty board, the game theoretic value is a first-player win. However, an explicit winning strategy is unclear — after an arbitrary opening move on the board, the theoretical result become unknown. Hex board can also be mapped into a Go-style grid, on which stones are played at intersections, as shown in Figure 1 (left). Usually 11×11 is considered as a regular board size; 13×13 is also played at computer Olympiad Hex tournaments.

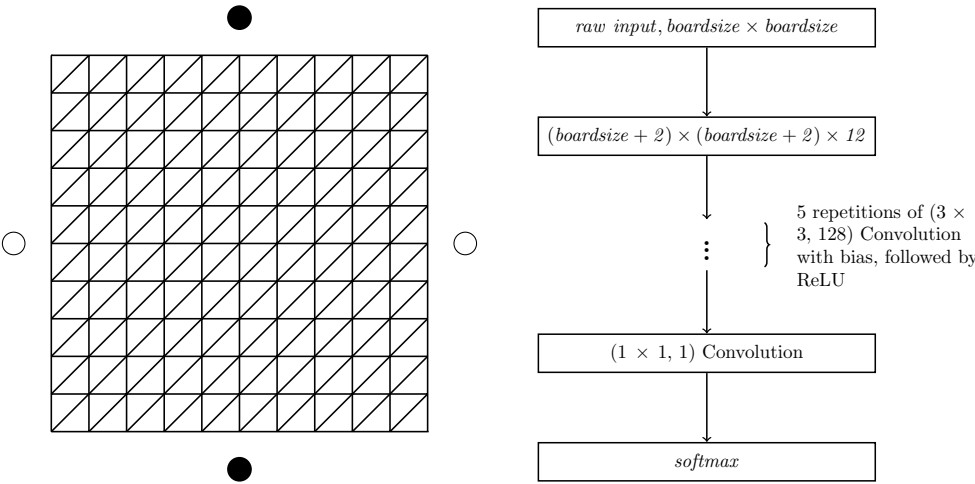

(a) A regular Hex board in Go-style representation. Stones are played at intersections. A player wins by connecting their two sides by a group of their stones.

(b) Neural network architecture: It accepts different board size inputs, padded with an extra border using black or white stones.

Figure 1: Hex board (left) and neural network design (right)

Hex is challenging to play primarily because of its large and near-uniform branching factor and the difficulty of constructing a reliable evaluation function. One character that makes Hex simpler is that perfect play can sometimes be achieved whenever winning virtual connections are found by H-Search (Anshelevich, 2002).

Similar to the previous work in Go, we use deep convolutional neural networks as a function approximator. Figure 1(right) shows our architecture. Instead of fixing the board size, we design an architecture that receives multiple board size inputs. This can be achieved by having multiple parallel inputs with different shape when building the computation graph using Tensorflow (Abadi et al., 2016). Our architecture allows board sizes $8 \leq N \leq 15$. After padding, each raw board state is processed into 12 binary feature planes, which are: black stones, white stones, empty points, to-play, black bridge endpoints, white bridge endpoints, to-play save bridge points, to-play make-connection points, to-play form bridge points, opponent save bridge points, opponent form bridge points, opponent make-connection points.

## 6.2 RESULTS

### 6.2.1 PURE NEURAL NET POLICY GRADIENT

We apply policy gradient reinforcement learning to improve a policy net which was trained by supervised learning. In addition to our proposed adversarial Monte carlo policy gradient algorithms, for comparison purposes, we implement three REINFORCE variants:

- REINFORCE-V: Vanilla REINFORCE using a parameterized self-play policy. After a batch of $n$ self-played games, each game is then replayed to determine batch policy gradient update $\frac{\alpha}{n} \sum_i^n \sum_t^{T_i} \nabla \log \pi(s_t^i, a_t^i; \theta) z_t^i$, where $z_t^i$ is either $+1$ or $-1$.
- REINFORCE-A: An "AlphaGo-like" REINFORCE. It differs from REINFORCE-V by randomly selecting a set of previously saved parameter weights from former iterations as the opponent for self-play.
- REINFORCE-B: For each self-played game, only one state-action pair is uniformly selected for policy gradient update. It differs from AMCPG-A by using the mean of all $k+1$ observed returns.

All methods are implemented using Tensorflow, sharing the same code base. They only differ in a few lines of code. A self-play policy is employed for all algorithms, which is equivalent to forcing $\pi_1$ and $\pi_2$ to share the same set of parameters. The game batch size $n$ is set to 128. For each self-play game, the opening move is played uniform randomly. Learning rate is set to 0.001 after a grid search, vanilla stochastic gradient ascent is used as the optimizer. The reward signal $z \in \{+1, -1\}$ only appears after a complete self-play game; no discounting is used. For all algorithms, the same neural net architecture is adopted. The initial parameter weights were obtained from supervised learning on a dataset generated by MoHex (Henderson, 2010) self-play on board size $9 \times 9$. This supervised learning is only on $9 \times 9$ Hex. Because there is no board-size dependent fully connected layers in our architecture, the parameter weights can be reused on other board sizes as the starting policy.

We run policy gradient reinforcement learning for 400 iterations for each method, on two different board sizes, $9 \times 9$ and $11 \times 11$, varying $k \in \{1, 3, 6, 9\}$. We use the improved Wolve (Henderson, 2010; Pawlewicz et al., 2015) as a benchmark to measure the relative performance of our learned models. After every 10 iterations, model weights are saved and then evaluated by playing against 1-ply Wolve. The primary strength of Wolve comes from its inferior cell analysis augmented H-Search, enabling it to discover winning strategies for perfect play much earlier than unenhanced H-Search. Wolve uses position evaluation with electric resistance evaluation (Anshelevich, 2002; Henderson, 2010; Pawlewicz et al., 2015) if no winning virtual connection can be found. Starting from trivial virtual connections (e.g., bridge pattern), H-Search (Anshelevich, 2002) computes larger virtual connections by iteratively applying an AND-OR combination rule, such a process is orders of magnitude slower than a pure neural net evaluation using GPU. The tournaments with Wolve are played by iterating all opening moves, each repeated 5 times with Wolve as black or white.

Figure 2 compares the strength of these five different algorithms. REINFORCE-B is able to achieve similar performance with REINFORCE-V, even though the number of training samples used in the

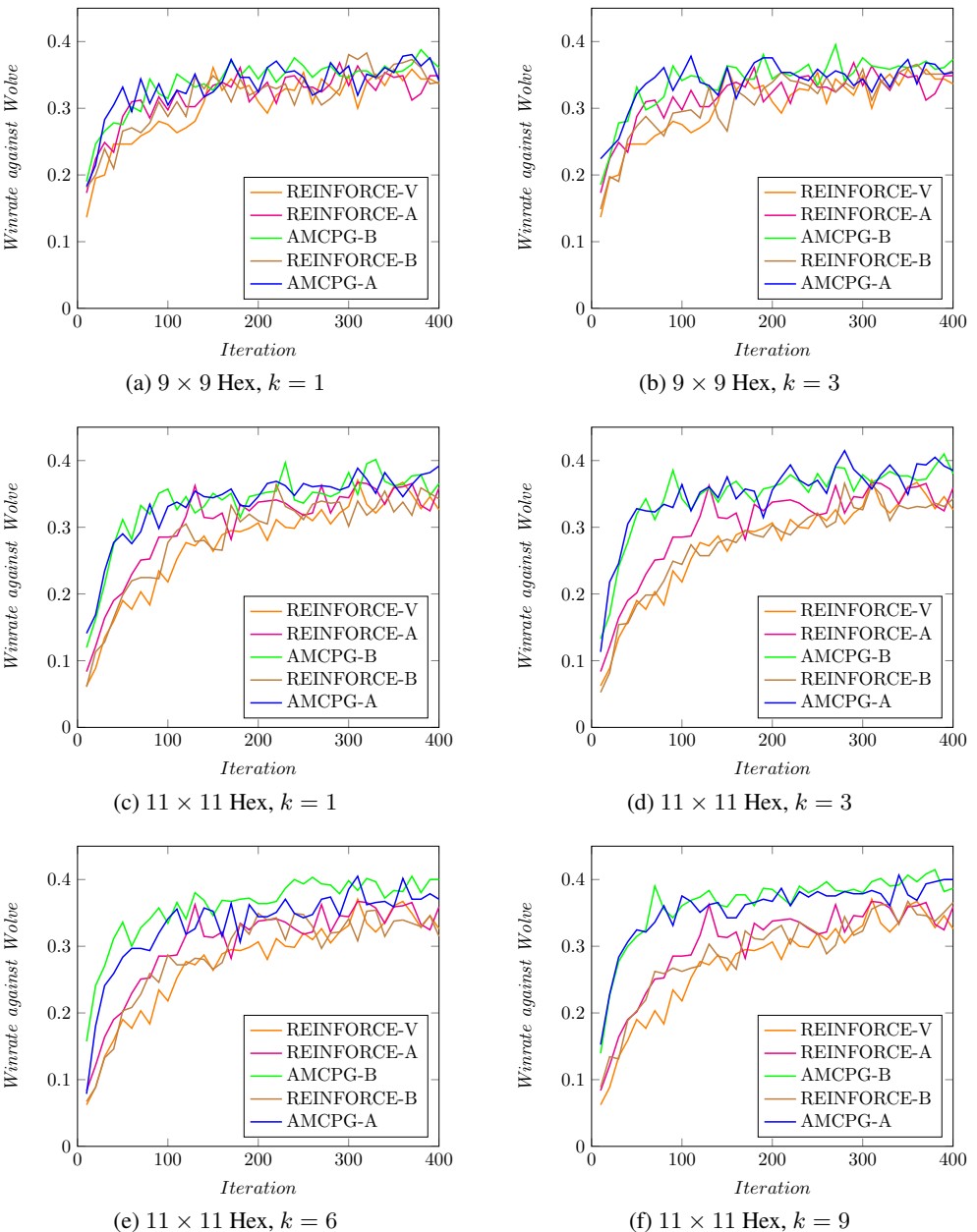

Figure 2: Comparison of playing strength against Wolve on $9 \times 9$ and $11 \times 11$ Hex with different $k$. The curves represent the average winrate among 10 trials with Wolve as black and white.

former algorithm is significantly less. This is perhaps because the reward signal in the same game is too much correlated. Consistent with the finding in Go (Silver et al., 2016), REINFORCE-A obtained small performance improvements on both $9 \times 9$ and $11 \times 11$ Hex. However, the newly developed algorithms AMCPG-A and AMCPG-B are able to learn better polices. They tend to learn faster, and generally achieve better results even when $k = 1$. The better performances are more clear on regular board size $11 \times 11$. Those results confirm the benefit of estimating the *minimum*, rather than *mean* return, when applying policy gradient methods to Alternating Markov Games.

The policy gradient training on both $9 \times 9$ and $11 \times 11$ Hex is very fast. On an Intel i7-6700 CPU computer with single GTX 1080 GPU, most took only a few hours. The longest training runs by AMCPG-A, AMCPG-B and REINFORCE-B ($k = 9$, $11 \times 11$ Hex) took about 12 hours. For $k = 3$ case, AMCPG-A, AMCPG-B and REINFORCE-B are about twice slower than REINFORCE-V and REINFORCE-A.

### 6.2.2 COMBINE NEURAL NET WITH MCTS FOR MULTIPLE BOARD SIZE HEX

Unlike previous work in Go (Silver et al., 2017b) or Hex (Anthony et al., 2017), the neural net we proposed has no board size dependent fully connected layers. This makes it possible to easily reuse the trained neural network on board sizes other than $9 \times 9$. To see the generalization ability across different board sizes in search, we incorporate the trained neural net to MoHex 2.0, the resulting program is MoHex-CNN$_9$, where we use subscript to reflect the nature that the CNN was optimized on $9 \times 9$ games and to distinguish it from another recently available player (Gao et al., 2017). The difference between MoHex-CNN$_9$ and MoHex 2.0 is that it replaces the prior probability of MoHex 2.0 with the policy neural net trained on $9 \times 9$ Hex games, i.e., the neural net model before policy gradient learning.

Table 1: Winrates of MoHex-CNN$_9$ against MoHex 2.0 with same number of simulations. All programs use the same default parameters as MoHex 2.0. Every opening was tried twice with each player as black or white. Column 6 and 7 are respectively the average consumed time per game for each corresponding program.

| Player | #Simulations | MoHex2.0 was white | MoHex2.0 was black | Overall winrate | Average time of MoHex-2.0 | Average time of MoHex-CNN | Board size |
|---|---|---|---|---|---|---|---|
| MoHex-CNN$_9$ | $10^4$ | 77.8% | 59.3% | 68.5% | 4.5 | 6.8 | $9 \times 9$ |
| MoHex-CNN$_9$ | $10^4$ | 83% | 58.0% | 70.5% | 12.3 | 17.1 | $10 \times 10$ |
| MoHex-CNN$_9$ | $10^4$ | 75.2% | 48.8% | 62.0% | 13.4 | 18.2 | $11 \times 11$ |
| MoHex-CNN$_9$ | $10^4$ | 78.5% | 55.6% | 67% | 29.4 | 36.0 | $12 \times 12$ |
| MoHex-CNN$_9$ | $10^4$ | 69.8% | 45.6% | 57.7% | 41.7 | 49.9 | $13 \times 13$ |

Table 1 shows the winrates of MoHex-CNN$_9$ on board sizes from $9 \times 9$ to $13 \times 13$. We note that board sizes smaller than 9 are not investigated because they can easily be solved by virtual connection computation or tree based proof number search (Henderson, 2010). MoHex 2.0 was not designed to play board size $> 13$, so we stop at board size 13. MoHex-CNN$_9$ defeats MoHex 2.0 on every board size, which is remarkable since the neural net model was only trained on $9 \times 9$ Hex games for a few hours on a single GPU. The computation overhead is not significantly big, because in our implementation, neural net evaluation is computed in parallel with *prior pruning* [1] when expanding a node. We note that when giving the same computation time per move, MoHex-CNN$_9$ could still beat MoHex 2.0, detailed results are shown in Appendix.

### 6.2.3 COMPARISON WITH EXIT

ExIt competed against MoHex 2011 (Arneson et al., 2010) on $9 \times 9$ Hex. To have a grasp of the relative strengths of ExIt, MoHex 2011, MoHex 2.0 and MoHex-CNN$_9$, we present results of MoHex 2.0 and MoHex-CNN$_9$ played against MoHex 2011 on $9 \times 9$ Hex [2]. We show the results with $10^4$ and $10^5$ simulations in Table 2. It should be noted that MoHex 2011 uses an expansion threshold of 1, while MoHex 2.0 and MoHex-CNN$_9$ adopt a threshold of 10, so when the same number

---

[1] Used by MoHex to prune proven inferior nodes whenever expanding a node

[2] In (Huang et al., 2013), MoHex 2.0 has been demonstrated to be stronger than MoHex 2011 on $11 \times 11$ and $13 \times 13$ Hex, but playing results on $9 \times 9$ board size were never presented.

of simulations are given, MoHex 2.0 and MoHex-CNN$_9$ expands much fewer nodes than MoHex 2011.

Table 2: Results of ExIt, MoHex 2.0 and MoHex-CNN$_9$ against MoHex 2011 on $9 \times 9$ Hex. All MoHex programs use the same default parameters, with $10^4$ simulations (left) or $10^5$ simulations (right). Every opening was tried twice with each player as black or white. ExIt used $10^4$ simulations (Anthony et al., 2017).

| Player | MoHex 2011 was white | MoHex 2011 was black | Overall winrate | MoHex 2011 was white | MoHex 2011 was black | Overall winrate |
|---|---|---|---|---|---|---|
| MoHex 2.0 | 82.7% | 66.7% | 74.7% | 79.0% | 65.4% | 72.2% |
| MoHex-CNN$_9$ | 85.2% | 65.4% | 75.3% | 84.0% | 71.6% | 77.8% |
| ExIt | - | - | 75.3% | - | - | 59.3% |

When $10^4$ simulations is used, both MoHex 2.0 and MoHex-CNN$_9$ has similar winrates as ExIt. However, the $10^5$ simulations MoHex 2.0 and MoHex-CNN$_9$ obtain much better winrates than 59.3%. Since ExIt was sticking on $10^4$ simulations, we conduct another experiments that use $10^4$ simulations MoHex-CNN$_9$ compete against MoHex 2011, in which setting, MoHex-CNN$_9$ won 62.3%, still better than ExIt, though MoHex-CNN$_9$'s node expansion is significantly less than that of MoHex 2011 and ExIt due to expansion threshold of 10 [3]. Indeed, as shown in Table 1, MoHex 2.0 and MoHex-CNN$_9$ consumes around 5s per game on $9\times9$ board size, indicating that with $10^4$ simulations per move they are playing super fast games. MoHex 2011's worse performance is not a surprise, under default settings, MoHex 2011 is even slighter weaker than Wolve (Hayward et al., 2012).

We further note that, the state-of-art for $9\times9$ Hex is that all openings have been solved (Pawlewicz & Hayward, 2013) [4]. This parallel solver has been embedded into MoHex 2.0, but in default setting, it is turned off.

## 7 CONCLUSIONS

We reviewed MDP and AMG, and proposed a new adversarial policy gradient paradigm that uses the worst-case return as the "critic". We introduced two practical adversarial Monte carlo policy gradient methods. We evaluated our algorithms on game of Hex, the experimental comparisons show that the new algorithms are notably better than routinely adopted self-play REINFORCE variants. We also applied *minimum* return to Monte carlo tree search, and observed better performance on $13\times13$ Hex. Another contribution we made is a multi-boardsize neural net architecture, we demonstrated that a single neural net model trained on smaller board size can effectively generalize to larger board sizes. As a result, using it as a prior knowledge, we are able to improve MoHex 2.0 consistently from $9\times9$ to $13\times13$ Hex. Since in practice, expert data is often scarce, difficult to generate with limited computation resources, in this paper we provided a more feasible approach for high-performance playing for multi-boardsize Hex.

On the other hand, based on the adversarial policy gradient formulation in AMGs, the Monte carlo policy gradients, either self-play REINFORCE-V or others, are not unbiased anymore, yet still have the drawback of being sample-inefficient, and may exhibit high variance. In this sense, an adversarial actor-critic framework is perhaps more appealing.

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

# APPENDIX

## A    PSEUDOCODE

We present the pseudocode of **AMCPG-A** and **AMCPG-B**.

---

**Algorithm 1:** Adversarial Monte-Carlo Policy Gradient (AMCPG-A and AMCPG-B)

---

1  Initialize $\theta$ ;
2  $ite \leftarrow 0$ ;
3  **while** $ite < maxIterations$ **do**
4  $\quad$ Self-play a batch of $n$ games $E$ using $\pi$ ;
5  $\quad$ **for** $e_i \in E$ **do**
6  $\quad\quad$ Select a state-action pair $(s_j, a_j)$ uniform randomly;
7  $\quad\quad$ Let $z(s_j, a_j)$ be the outcome with respect to action $a_j$ at $s_j$ in $e_i$;
8  $\quad\quad$ From $(s_j, a_j)$,

$$z'(s_j, a_j) = \begin{cases} A: & \text{Self-play } k \text{ games using } \pi, \text{ record the minimum outcome with respect to } s_j; \\ B: & \text{Select top } k \text{ moves from } s'_j, \text{ from each move self-play a game using } \pi, \\ & \text{record the minimum outcome w.r.t } s_j; \end{cases}$$

$\quad\quad\quad R^i \leftarrow \min\{z(s_j, a_j), z'(s_j, a_j)\}$ ;
9  $\quad\quad$ Write $(s_j, a_j, R^i)$ to the batch ;
10 $\quad$ $\theta \leftarrow \theta + \frac{\alpha}{n} \sum_{i=1}^{n} \nabla \log \pi(s_j^i, a_j^i; \theta) R^i$;
11 $\quad$ $ite \leftarrow ite + 1$;

---

We also present the pseudo-code of **Algo.1**, **Algo.2** and **Algo.3** in Section 3.

## B    EXPERIMENTS

### B.1    SUPERVISED LEARNING

Since high quality data is not instantly available, to initialize the neural net weight, we first generated a dataset containing $10^6$ state-action pairs by computer player MoHex played against MoHex, on $9 \times 9$ board. We train the policy neural net to maximize the log-likelihood on this dataset. The training took $100,000$ steps with mini-batch 64, optimized using Adam (Kingma & Ba, 2015) with learning rate 0.001.

The resulting neural net could be used for multiple board sizes, its win-percentages against 1-play Wolve on $9 \times 9$ and regular board size $11 \times 11$ are respectively $13.2\%$ and $4.6\%$ (iterated all opening moves, 10 trails each opening with Wolve as Black and White).

Input feature has 12 binary planes, utilizing a "bridge pattern" in Hex.

### B.2    MOHEX 2.0's IN-TREE PHASE

Starting from the root node, MoHex 2.0 selects a child node according to the score function in below:

$$score(s,a) = (1-w)\left(q(s,a) + c_b\sqrt{\frac{\log N(s)}{N(s,a)}}\right) + wR(s,a) + c_{pb}\frac{\rho(s,a)}{\sqrt{N(s,a)+1}},$$

where $N(s), N(s,a)$ are respectively the visit counts of $s$ and $(s,a)$, $q(s,a)$ is the Q-value of $(s,a)$, $R(s,a)$ is the RAVE value, $\rho(s,a)$ is the prior probability from move pattern weights, $w, c_b, c_{pb}$ are weighting parameters. MoHex-CNN uses the same formula except that the prior probability is from neural net.

---

**Algorithm 2:** Policy Iteration algorithms for AMGs

---

1  **Procedure** `Algo.1()`:
2      $\pi_1, \pi_2$ are policies for player 1 and player 2;
3      Initialize $\pi_1^0, \pi_2^0$ arbitrarily;
4      $t \leftarrow 1$;
5      $\pi_1^t \leftarrow \pi_1^0$;
6      $stable \leftarrow false$;
7      **while** *not stable* **do**
8          Fix $\pi_1^t$;
9          Compute the optimal counter strategy for player 2 given player 1's stragy is fixed;
10         Let $\pi_2^t$ be the computed optimal counter strategy;
11         Fix $\pi_2^t$;
12         Compute the optimal counter strategy for player 1 given player 2's strateg is fixed;
13         Let $\pi_1^{t+1}$ be the computed optimal counter strategy;
14         $t \leftarrow t + 1$;
15         **if** $\pi_1^{t-1} = \pi_1^t$ *and* $\pi_2^{t-2} = \pi_2^{t-1}$ **then**
16             $stable \leftarrow true$;

17 **Procedure** `Algo.2()`:
18     $\pi_1, \pi_2$ are policies for player 1 and player 2;
19     Suppose player 1 is the max player, palyer 2 is the min player;
20     Initialize $\pi_1^0, \pi_2^0$ arbitrarily;
21     $t \leftarrow 0$;
22     $stable \leftarrow false$;
23     **while** *not stable* **do**
24         Fix $\pi_1^t, \pi_2^t$;
25         Compute the value function $V_t$ under $\pi_1^t, \pi_2^t$ by policy evaluation;
26         Let $\pi_1^{t+1}$ be the greedily (max) modified policy w.r.t $V_t$;
27         Let $\pi_2^{t+1}$ be the greedily (min) modified policy w.r.t $V_t$;
28         $t \leftarrow t + 1$;
29         **if** $\pi_1^t = \pi_1^{t-1}$ *and* $\pi_2^t = \pi_2^{t-1}$ **then**
30             $stable \leftarrow true$;

31 **Procedure** `Algo.3()`:
32     $\pi_1, \pi_2$ are policies for player 1 and player 2;
33     Suppose player 1 is the max player, palyer 2 is the min player;
34     Initialize $\pi_1^0, \pi_2^0$ arbitrarily;
35     $t \leftarrow 0$;
36     $stable \leftarrow false$;
37     **while** *not stable* **do**
38         Fix $\pi_1^t, \pi_2^t$;
39         Compute the value function $V_t$ under $\pi_1^t, \pi_2^t$ by policy evaluation;
40         Let $\pi_1^{t+1}$ be the greedily (max) modified policy w.r.t $V_t$;
41         Fix $\pi_1^{t+1}$;
42         Compute the optimal counter policy for player 2 given player 1's policy is fixed;
43         Let $\pi_2^{t+1}$ be this optimal counter policy;
44         $t \leftarrow t + 1$;
45         **if** $\pi_1^t = \pi_1^{t-1}$ *and* $\pi_2^t = \pi_2^{t-1}$ **then**
46             $stable \leftarrow true$;

---

| Plane index | Description | Plane index | Description |
|:---:|:---:|:---:|:---:|
| 0 | Black played stones | 6 | To play save bridge points |
| 1 | White played stones | 7 | To play make-connection points |
| 2 | Unoccupied points | 8 | To play form bridge points |
| 3 | Black or White to play | 9 | Opponent's save bridge points |
| 4 | Black bridge endpoints | 10 | Opponent's form bridge points |
| 5 | White bridge endpoints | 11 | Opponent's make-connection points |

### B.3 WITH THE SAME COMPUTATION TIME AND DFPN SOLVER TURNED ON

In computer Hex Olympiad tournament, MoHex is usually played with the parallel solver turned on. To demonstrate the full strength of MoHex-CNN$_9$ compared to MoHex 2.0, we perform another experiments by limiting both programs with the same thinking time 5s per move, all with the DFPN solver turned on.

Table 3: Winrates of MoHex-CNN$_9$ against Mohex 2.0 with the same time per move. All programs use the default parameters the same as MoHex 2.0 with parallel solver on. Every opening was tried twice with each player as black or white.

| Opponent | Time per move (s) | MoHex2.0 was white | MoHex2.0 was black | Overall winrate | Average length of played games | Board size |
|:---|:---:|:---:|:---:|:---:|:---:|:---:|
| MoHex-CNN$_9$ | 5 | 66.7% | 51.9% | 59.3% | 39.5 | 9×9 |
| MoHex-CNN$_9$ | 5 | 64.0% | 56.0% | 60.0% | 48.2 | 10×10 |
| MoHex-CNN$_9$ | 5 | 68.6% | 56.2% | 62.4% | 58.1 | 11×11 |
| MoHex-CNN$_9$ | 5 | 64.0% | 52.8% | 59.4% | 69.7 | 12×12 |
| MoHex-CNN$_9$ | 5 | 57.4% | 55.6% | 56.5% | 80.6 | 13×13 |

As expected, the results show that on smaller board size like $9 \times 9$, due to the DFPN solver in both players, there is a significant decrease of MoHex-CNN$_9$'s winrate. However, on larger board sizes, the influence of DFPN solver is small. We note that, in (Anthony et al., 2017), on 9×9 board size, ExIt's winrate against 4s-MoHex2011 with DFPN solver is 55.6%. MoHex 2.0's solver is also stronger than MoHex 2011 due to improved virtual connections (Pawlewicz et al., 2015).

