# OpenReview forum: "Adversarial Policy Gradient for Alternating Markov Games"
_ICLR.cc/2018/Conference — Invite to Workshop Track_

### Official Review · AnonReviewer1 · 2017-11-23
**n/a**

**Rating:** 5
**Confidence:** 2

**Review:**

This paper is outside of my area of expertise, so I'll just provide a light review:

- the idea of assuming that the opponent will take the worst possible action is reasonable in widely used in classic search, so making value functions follow this intuition seems sensible,
- but somehow I wonder if this is really novel? Isn't there a whole body of literature on fictitious self-play, including need RL variants (e.g. Heinrich&Silver, 2016) that approaches things in a similar way?
- the results on Hex have some signal, but I don’t know how to calibrate them w.r.t. The state of the art on that game? A 40% win rate seems low, what do other published papers based on RL or search achieve?

---

> ### Author Response · Authors · 2017-12-21
> **New results after combing neural net with search have been added to the paper**
>
> Thanks for the reviewing.
>
> The reviewer mentioned fictitious self-play (Heinrich&Silver, 2016), but it is primary for imperfect-information games.
>
> We focus on classic perfect information two-player zero-sum game played in alternate turns.
>
> Additionally, the reviewer was concerned about the state-of-art in Hex. In the revised paper, we haven shown that after combining our neural net with search, the state-of-art in Hex is improved. Moreover, we used a single neural net model, with consistent improvement on multiple board sizes.

---

### Official Review · AnonReviewer2 · 2017-11-27
**A bit verbose on existing methods + notations and low on experiments**

**Rating:** 5
**Confidence:** 4

**Review:**

This paper introduces a variation over existing policy gradient methods for two players zero sum games, in which instead of using the outcome of a single policy network rollout as the return, they use the minimum outcome among a few rollouts either from the original position or where the first action from that position is selected uniformly among the top k policy outputs.

The proposed method supposedly provides slightly stronger targets, due to the extra lookahead / rollouts. Experiments show that this provides faster progress per iteration on the game of Hex against a fixed third party opponent.

There is no comparison against state of the art methods like AlphaGo Zero which uses MCTS root move distribution and MCTS rollouts outcome to train policy and value network, even though the author do cite this work. There is also no comparison with Hexit which also trains policy net on MCTS move distribution, and was also applied to Hex.

The actual proposed method is actually a one liner change, which could be introduced much sooner in the paper to save the reader some time. While the idea is interesting, the paper felt quite verbose on introducing notations and related work, and a bit lacking on actual change that is being proposed and the experiment to back it up.

For example,  was it really necessary to introduce state transition probabilities p(s’, a, s) when all the experiments are done in the deterministic game of Hex ?

Also the experiment seems not fully fair to the reinforce baseline. My understand is that the proposed method is much more costly due to extra rollouts that are needed. It would be interesting to see the same learning curves as in Figure 2, but the x axis would be some computational budget (total number of network forward, or wall clock time). It is conceivable that the vanilla reinforce would do just as well as the proposed method if the plots were aligned this way. It would also be good to know the asymptotic behavior.

So even though the idea is interesting, it seems that much stronger methods AlphaGo Zero / Hexit are now available, and the experimental section is a bit weak. I would recommend to accept for a workshop paper but not sure about the main track.

---

> ### Author Response · Authors · 2017-12-03
> **Comparison to ExIt has been added**
>
> Thanks for your comment.
>
> In the revised paper, we have added our neural net model to search, the resulting program is stronger than MoHex 2.0 on board sizes 9x9 to 13x13.  We have also included a comparison with ExIt. It appears that ExIt might not as strong as MoHex 2.0 (the ExIt paper was comparing their player with MoHex 2011).  Another advantage of our new player is that it is able to play on multiple board size with only one trained model, while ExIt is limited on 9x9.
>
> Detained responses are in below.
>
> The methods ExIt (I assume you mean ExIt by saying HexIt) and AlphaGo Zero are similar. They work well but one problem is the computation cost is very high. For example, when applied to chess and shogi, it is mentioned that 5000 TPUs were used for MCTS self-play data generation.
>
> For ExIt, by the time our paper is submitted, only first version is available on arxiv, though we are aware their work has been accepted in NIPS 2017.  The newest version can be found from this URL.
>  https://arxiv.org/abs/1705.08439
>
> They did all experiments on 9x9 Hex. In the first version on arxiv,  their player is a search+NN player not pure neural net.
> On other hand hand, even if the learned neural net policy itself is strong by following MCTS, it is likely the playing strength of this pure neural net can be improved by doing a policy gradient on it, though after such a policy gradient, the policy might not good for Monte-carlo tree search any more (as shown by first Alphago paper).
>
> In the newest version, they compared their policy_value net + MCTS player with MoHex 2011, however, there is MoHex 2.0, which is much stronger than MoHex 2011.
>
> ExIt only conducts experiments on 9x9 Hex. It is not very clear how much time could be used to produce significant results on larger board size, such as 11x11, presumably, this is not a easy task with only one GPU computer. We note that even ExIt was specially applied only to this board size, MoHex 2.0 and our new program both seem to be able achieve better playing results than ExIt.
>
> Our AMCPG-A or AMCPG-B follows traditional “light self-play”. No tree was built. To estimate the “minimum” critic, extra roll-outs are conducted. But it is very much due to the Monte-Calro nature of the method, and that is why we mention an actor-critic style might be more efficient. Our methods work essentially similar as traditional policy gradient, that's why we only compared with REINFORCE variants.
>
> We argue that it could be unfair to say that our better results compared to classic REINFORCE is merely due to extra roll-outs.  One can see that in REINFORCE-B, extra roll-outs are also conducted the same way as AMCPG-A and AMCPG-B.  Their extra computation costs due to extra roll-out are the same.  However, the results in Figure 2 suggests that REINFORCE-B has similar performance as REINFORCE-A and REINFORCE-V.

---

> ### Author Response · Authors · 2017-12-03
> **computation cost**
>
> Thanks for your comments.
>
> The reviewer is concerned about the computation cost for each training. In our experiments, training is very fast, took only a few hours on 9x9 and 11x11 Hex. Since all training/evaluation are conducted on the same computer with a single GTX 1080 GPU. We briefly list the detailed training time for each method here:
>
> 9x9 Hex:  total time usage for 400 iterations training:
> AMCPG-A: k=1: about 1 h 40 m,  k=3: about 2.5 h,  k=6: 4h 10 minutes, k=9: about 6h
> AMCPG-B:  similar as above
> REINFORCE-B:  similar as above
> REINFORCE-A: 1 hour 15 minutes
> REINFORCE-V: 1 hour 20 minutes
>
> 11x11 Hex:
> AMCPG-A: k=1: about 3h15 minutes, k=3: about 5h, k=6: about 9h, k=9: about 12h
> AMCPG-B: similar as above
> REINFORCE-B:  similar as above,
> REINFORCE-A: 2.5 hours
> REINFORCE-V: 2.5 hours
>
> In fact, most of our time was not spending on training the neural net, but for evaluation the neural net model by playing against Wolve, as Wolve's search is orders of magnitude slower than a pure neural net player.
>
> We note that, even though a pure neural net self-play training might not be able to provide state-of-the-art playing, such methods have its own merits.  For example, due to their fast speed, the first version of Alphago uses such a method for generating data to train a value net which is useful in search.
>
> On other hand hand, even though search+NN self-play might also be able to learn a neural net policy that itself can play strongly. It is likely that such a neural net could be further improved by  policy gradient.

---

### Official Review · AnonReviewer3 · 2017-11-27
**A nice but somewhat minimal paper addressing caveats of existing adversarial RL attempts**

**Rating:** 5
**Confidence:** 4

**Review:**

The paper makes the simple but important observation that (deep) reinforcement learning in alternating Markov games requires a min-max formulation of the Bellman equation as well as careful attention to the way in which one alternates solving for both players' policies in a policy iteration setting.

While some of the core algorithmic insights regarding Algorithms 3 & 4 in the paper stem from previous work (Condon, 1990; Hoffman & Karp, 1966), I was not actually aware of these previous results until I reviewed this paper.

A nice corollary of Algorithms 3 & 4 is that they make for a straightforward adaptation of policy gradient algorithms since when optimizing one policy, the other is fixed to the greedy policy.

In general, it would be nice to have the algorithms specified as formal algorithms as opposed to text-based outlines.  I found myself reading and re-reading descriptions to make sure I understood what math was being implied by the descriptions.

Section 6

> Hex is simpler than Go in the sense that perfect play can
> often be achieved whenever virtual connections are found
> by H-Search

It is not clear here what virtual connections are, what H-Search is, and how these imply perfect play, if perfect play as previously discussed is unknown.

Overall, the results on Hex for AMCPG-A and AMCPG-B vs. standard REINFORCE variants currently used are very encouraging.  That said, empirically it is always a question of whether these results are specific to Hex.  Because this paper is not proposing the best Hex player (i.e., the winning rate against Wolve never exceeds 0.5), I think it is quite reasonable to request the authors to compare AMCPG-A and AMCPG-B to standard REINFORCE variants on other games (they do not need to be as difficult as Hex).

Finally, assuming that the results do generalize to other games, I am left wondering about the significance of the contribution.  On one hand, the authors have introduced me to literature I was not aware of, but on the other hand, their actual novel contribution is a rather straightforward adaptation of ideas in the literature to policy gradients (that could be formalized in a more technically precise way) with an evaluation on a single type of game.  This is a useful contribution no doubt, but I am concerned with whether it meets the significance level that I am used to with accepted ICLR papers in previous years.

---

> ### Author Response · Authors · 2017-12-21
> **previous discussion about AMGs was "neglected"**
>
> Thank you for your comments.
>
> Yes, the key insights behind this paper is much from the literature, i.e., (Condon, 1990; Hoffman & Karp, 1966; Littman 1996).  But, as the reviewer has pointed out, perhaps it is because of the difference in terminology, those classic works were much "unknown" for many researchers.
>
> In this paper, we brought those again to the community, one goal is to stimulate more thorough thinking about the difference between two-player alternate-turn games and single agent MDPs.  It is apparent that two-player alternate-turn zero-sum games are more "challenging" in many aspects. A more careful examination about the fundamental differences between AMGs and MDPs will perhaps help people develop more effective/efficient RL methods specifically for this domain.
>
> We only did our experiments on the game of Hex, primarily because this is the game we are most familiar. But it should be noted that we didn't conduct any game specific modifications when applying those AMCPG variants to this specific game, just as  REINFORCE.
>
> It is true that doing more games would be more convincing; however, due to various constraint (i.e., hardware constraint, knowledge about other games), we did not manage to have an attempt in this direction while writing this paper.
>
> As for advancing the state-of-art,  the state-of-the-art for Hex are still search based methods. In the first version we submitted, we did not attempt to advance the state-of-art, since we were concentrated on introducing new fast and better policy gradient methods.
>
> However, after receiving the reviewers' comments about state-of-art, we proceed to combine our neural net with search, and the resulting program is indeed be able to surpass MoHex 2.0.
>
> Most notably, we use a single model for multiple board sizes, the new program consistently defeats MoHex 2.0 on every board size.  This is much due to the architecture we introduced, where we deliberately removed fully connected layers, so that the learned parameter weights can generalize to multiple board sizes.
>
> Since expert data is often difficult to obtain or generate, while generating expert data on smaller board is usually much easier and cheaper than larger board sizes, our result provides an encouraging direction for more efficient learning on games which has similar characteristics as Hex (e.g., other connection games).
>
> We have also investigated “minimum return” in Monte-carlo tree search, experimental results show that incorporating “minimum playout” also improved MCTS.
>
> Future work direction is using value net in pure neural net training as well as use it to replace the playout in MCTS.  However, different from previous work, we argue that a “min” operator might be able to lead better results in alternating markov games.
>
> We have included a psude-code for Algo.1, Algo.2 and Algo.3 in the appendix, which provides a more formal discription about each procedure. Also, explanation about Virtual Connections and H-Search have been added  in the revised paper.

---

> > ### Comment · AnonReviewer3 · 2018-01-10
> > **Response**
> >
> > Overall, I like the paper (it makes a simple but important point) and the authors have addressed most of my concerns.
> >
> > That said, the one major issue that remains with the paper is that I would like to see evaluations in a larger variety of domains -- I feel like I'm overfitting my understanding of the ideas in the paper to the game of Hex.  For this reason, I feel that my current review score is appropriate.  As another reviewer points out, this paper would be great for a workshop if it is not accepted to the main track.

---

### Author Response · Authors · 2017-12-18
**a new version has been updated**


We thank the reviewer's comments about the 'state-of-the-art' in Hex.  We have updated our paper, in which we show that after combining our neural net model with search, better results than MoHex 2.0 are observed.

We summarize the changes in below:


1. we show that with our boardsize independent (as there is no fully connected layers) neural net architecture, a single trained model trained on 9x9 can generalize to other board sizes. When combined with search, the new program consistently defeats MoHex 2.0 on 9x9 to 13x13 (with same number of simulations and with same computation time ).

2. we also compared our results with ExIt. We show that both MoHex 2.0 and our new program achieve better winrates against MoHex 2011 than ExIt, though ExIt was only concentrated on 9x9.

3.  we show that minimum rollout return slightly improves Monte carlo tree search


4. typos and grammatical errors are corrected.

However, due to various constraint, we did not apply our methods to other games, though it would be interesting to do so. Hex is the game we are most familiar. But on the other hand,  we stress that, just as REINFORCE, we did not have any special modifications when apply the ACMPG variants to Hex.

---

### Decision · Program_Chairs · 2018-01-29
**ICLR 2018 Conference Acceptance Decision**

**Decision:**

Invite to Workshop Track

**Comment:**

The reviewers agree that the paper is below threshold for acceptance in the main track (one with very low confidence), but they favor submitting the paper to the workshop track.

The paper considers policy gradient methods for two-player zero-sum Alternating Markov games.  They propose adversarial policy gradient (fairly obviously), wherein the critic estimates min rather than mean reward.   They also report promising empirical results in the game of Hex, with varying board sizes.  I found the paper to be well-written and easy to read, possibly due to revisions in the rebuttal discussions.

The reviewers consider the contribution to be small, mainly due to the fact that the key algorithmic insights were already published decades ago.  Reintroducing them is a service to the community, but its novelty is limited.  Other critiques mentioned that results in Hex only provide limited understanding of the algorithm's behavior in general Alternating Markov games.  The lack of comparison with modern methods like AlphaGo Zero was also mentioned as a limitation.

Bottom line: The paper provides a small but useful contribution to the community, as described above, and the committee recommends it for workshop.